# A Segmental Approach from Molecular Profiling to Medical Imaging to Study Bicuspid Aortic Valve Aortopathy

**DOI:** 10.3390/cells11233721

**Published:** 2022-11-22

**Authors:** Froso Sophocleous, Estefania De Garate, Maria Giulia Bigotti, Maryam Anwar, Eva Jover, Aranzazu Chamorro-Jorganes, Cha Rajakaruna, Konstantina Mitrousi, Viola De Francesco, Aileen Wilson, Serban Stoica, Andrew Parry, Umberto Benedetto, Pierpaolo Chivasso, Frances Gill, Mark C. K. Hamilton, Chiara Bucciarelli-Ducci, Massimo Caputo, Costanza Emanueli, Giovanni Biglino

**Affiliations:** 1Bristol Medical School, Faculty of Health Sciences, University of Bristol, Bristol BS8 1TH, UK; 2Bristol Heart Institute, University Hospitals Bristol and Weston NHS Foundation Trust, Bristol BS1 3NU, UK; 3School of Biochemistry, Faculty of Life Sciences, University of Bristol, Bristol BS8 1TH, UK; 4National Heart and Lung Institute, Imperial College London, London SW7 2BX, UK; 5Instituto de Investigación Sanitaria de Navarra, 31008 Pamplona, Spain; 6Department of Clinical Radiology, University Hospitals Bristol, Bristol Royal Infirmary, Bristol BS2 8EJ, UK; 7Royal Brompton & Harefield Hospitals, Guy’s and St Thomas’ NHS Foundation Trust, London SW3 6NP, UK

**Keywords:** aortic segmentation, microRNAs, proteins, wall shear stress

## Abstract

Bicuspid aortic valve (BAV) patients develop ascending aortic (AAo) dilation. The pathogenesis of BAV aortopathy (genetic vs. haemodynamic) remains unclear. This study aims to identify regional changes around the AAo wall in BAV patients with aortopathy, integrating molecular data and clinical imaging. BAV patients with aortopathy (n = 15) were prospectively recruited to surgically collect aortic tissue and measure molecular markers across the AAo circumference. Dilated (anterior/right) vs. non-dilated (posterior/left) circumferential segments were profiled for whole-genomic microRNAs (next-generation RNA sequencing, miRCURY LNA PCR), protein content (tandem mass spectrometry), and elastin fragmentation and degeneration (histomorphometric analysis). Integrated bioinformatic analyses of RNA sequencing and proteomic datasets identified five microRNAs (miR-128-3p, miR-210-3p, miR-150-5p, miR-199b-5p, and miR-21-5p) differentially expressed across the AAo circumference. Among them, three miRNAs (miR-128-3p, miR-150-5p, and miR-199b-5p) were predicted to have an effect on eight common target genes, whose expression was dysregulated, according to proteomic analyses, and involved in the vascular-endothelial growth-factor signalling, Hippo signalling, and arachidonic acid pathways. Decreased elastic fibre levels and elastic layer thickness were observed in the dilated segments. Additionally, in a subset of patients n = 6/15, a four-dimensional cardiac magnetic resonance (CMR) scan was performed. Interestingly, an increase in wall shear stress (WSS) was observed at the anterior/right wall segments, concomitantly with the differentially expressed miRNAs and decreased elastic fibres. This study identified new miRNAs involved in the BAV aortic wall and revealed the concomitant expressional dysregulation of miRNAs, proteins, and elastic fibres on the anterior/right wall in dilated BAV patients, corresponding to regions of elevated WSS.

## 1. Introduction

Bicuspid aortic valve (BAV) is the most common congenital heart disease with 1–2% prevalence in the general population, and its heterogenous nature commonly affects both the aortic valve and the aorta, leading to aortic valve stenosis/regurgitation and progressive ascending aortic (AAo) dilation [1,2]. The association between AAo dilation and BAV has been thoroughly confirmed, yet the exact molecular and haemodynamic mechanisms leading to BAV-associated aortopathy remain unclear [3].

MicroRNAs are small non-coding RNAs, which canonically repress the expression of their “target gene” at the post-transcriptional level, and are abundantly expressed in the vasculature, guiding vascular development and remodelling [4]. They have been implicated in aneurysm formation in the ascending aorta [5] and have been proposed as circulating biomarkers in this clinical setting [6]. Increasing evidence indicates that miRNAs play a role in mechanotransduction events [7], suggesting that structural alterations in aortopathy could be partially mediated by miRNAs sensitive to cyclic biomechanical stress (“mechanomiRs”).

We therefore conducted a detailed prospective study in a population of patients with BAV aortopathy, collecting tissue biopsies and imaging data to study local, segmental molecular, and haemodynamic changes around the aortic circumference. This study proposes a methodological framework to conduct segmental analyses on different molecular levels, complemented by additional preliminary haemodynamic observations, to study the complex nature of BAV aortopathy through dynamic and likely intertwined changes. In the literature, two studies [8,9] compared media degeneration, including elastic degradation to wall shear stress (WSS), while another study [10] compared miRNA changes between the inner and outer curvature of the AAo. Combining miRNA and proteomic analyses with observations on histomorphometric and haemodynamic components across different segments of the aortic circumference is promising in the unravelling of the complex mechanisms underlying BAV aortopathy.

The overarching aim of this study was to propose an approach of aortic segmentation to generate new knowledge on the mechanisms underlying aortic dilation in BAV patients, derived from comparisons of the most likely vs. the least likely aortic segments to be involved. The specific objectives were: (a) to identify and validate changes in miRNA and protein expression between dilated and non-dilated aortic segments; (b) to bioinformatically identify the association between dysregulated miRNAs and proteins and the wider molecular pathways they could control; (c) to measure the amount and thickness of elastin in the same aortic segments as a marker of adverse remodelling due to aortopathy; and (d) to measure WSS in the AAo and elucidate links between dysregulated miRNAs and WSS levels. 

## 2. Materials and Methods

### 2.1. Patient Population

Patients scheduled for aortic replacement (i.e., Ross procedure, interposition aortic graft/bypass, aortic valve sparing, Ozaki procedure, and elephant trunk/whole arch replacement) were identified for biopsy collection based on their valve morphology (i.e., bicuspid) and invited for a pre-surgical research cardiovascular magnetic resonance (CMR) scan. The decision for aortic replacement followed ACC/AHA Guidelines 2020 [11]. Exclusion criteria are reported in the Appendix B. Informed consent for the research use of images and the use of waste aortic tissue collected during surgery was obtained from the participating patients. The study received Health Research Authority approval following the Research Ethics Committee (REC) review (REC references: 17/NI/0147, 15/LO/1064). The investigation of human tissue conforms to the principles outlined in the Declaration of Helsinki.

### 2.2. Data Collection

Aortic segments were collected at the time of surgery and cut into six segments (Figure 1), namely the anterior (A), anterior-right (AR), posterior-right (PR), posterior (P), posterior-left (PL), and anterior-left (AL). Each segment was further divided into three pieces to be stored as follows: (i) one piece in 500 μL RNA later for miRNA analysis, stored at −80 °C; (ii) a second piece flash-frozen for proteomic analysis, stored at −80 °C; and (iii) a third piece in 4 mL of 0.1M phosphate-buffered saline (PBS) to be subsequently fixed in a 4% formaldehyde solution and embedded in paraffin wax for histomorphometric analysis. 

When possible and when patients consented to the additional imaging, research CMR scans were performed at 3T (Magnetom Skyra, Siemens Healthineers, Erlangen, Germany), including an aortic 4D flow sequence for haemodynamic assessment.

Demographic (i.e., sex, age at time of study, and body surface area (BSA)) and clinical data were collected from the patients’ clinical records (see Appendix A). 

### 2.3. Molecular Analyses

#### 2.3.1. Next-Generation RNA Sequencing and Automated miRCURY LNA miRNA PCR Assay

Aortic tissue (20–30 mg) preserved in RNA was later homogenised in 700 μL QIAzol Lysis Reagent with 2.8 mm ceramic beads, using a Bertin homogeniser at 4 °C. An MiRNeasy Mini Kit was used for total miRNA extraction (No. 217004, Qiagen, Hilden, Germany), which was performed for n = 8 patients (Table 1: P2, 3, 6, 7, 8, 9, 14, and 15). As per clinical assessment, the anterior-right wall is dilated in the case of asymmetric BAV dilation; thus, two opposite segments per patient were selected, i.e., AR (side of dilation) vs. PL (diametrically opposite anatomical side). These patients were selected based on the presence of asymmetric (AR) dilation, no severe mix valve disease, and no severe calcification, and an equal number of males and females were selected to avoid gender bias. The RNA extract was sent to Qiagen for NGS analysis and selected markers were then validated by an automated miRCURY LNA miRNA PCR assay (Qiagen) [12] by expanding the initial cohort to n = 15 patients, including an available n = 3 patients with overall aortic dilation (rather than one-sided, asymmetrical) for comparison. Additional patients are also listed in Table 1. This analysis was performed on all 6 aortic segments per patient to explore the full spectrum of anatomical changes; therefore, a combination of adjacent segments, i.e., A + AR vs. P + PL, was also included to validate results on the most involved vs. the least involved segments and to extend the observations, taking into account possible human error in segmentation. The miRNA data analyses in the Appendix A show the calculations used. 

#### 2.3.2. Mass-Spectrometry Analysis

Flash-frozen aortic tissue (20–30 mg) from the same n = 8 patients chosen for NGS (two segments per patient, i.e., AR vs. PL) was homogenised, RIPA-buffered (i.e., using gentleMACS Dissociator), and centrifuged to collect the supernatant. The homogenised samples were shaken on ice for 30 min and then centrifuged at 13,000× *g* for 10 min at 4 °C. The aliquoted supernatant was sent to the University of Bristol Proteomics Facility for tandem mass spectrometric analysis. 

### 2.4. Bioinformatic Analysis

Using miRWalk suite (v.2.0, University of Heidelberg, Heidelberg, Germany,) and selecting the TargetScan and miRDB databases, as well as a confidence score of 0.90, target prediction was carried out for the validated miRNA markers (which were significantly dysregulated between the anatomically opposite aortic segments, i.e., AR vs. PL, considering adjacent segments, thus comparing A + AR vs. P + PL). The predicted targets were then compared to significantly dysregulated proteins derived from two analyses of the same patients. Downregulated gene targets were selected for upregulated miRNAs, and vice versa. Gene ontology and protein pathways were extracted from Webgestalt for gene targets with >0.5 and <−0.5 log_2_ (fold change).

### 2.5. Histomorphometric Analysis

Paraffin-embedded samples for the same n = 8 patients (two segments per patient, i.e., AR vs. PL) were cut on a microtome (5 μm thickness) and stained with elastin van Gieson (EVG) using an autostainer machine. Microphotograph imaging and analyses were performed blinded to the clinical diagnosis. Two serial sections of each tissue were prepared to obtain technical duplicates per tissue sample, and 8 arbitrary fields were imaged per section at 40 and 200× magnifications to analyse both the thickness of the elastic layer and the content of the elastic fibres, in the intima and the media layers of the aortic wall, respectively. The contents of the elastic fibres were quantified using Image J software (National Institutes of Health, Bethesda, MD, USA). The thickness was analysed by averaging the length of 5 transversal sites of the area comprised of elastic fibres.

### 2.6. Haemodynamic Analysis

Haemodynamic parameters were analysed using Caas MR 4D flow software (v5.2.1, Pie Medical Imaging, Maastricht, The Netherlands). The magnitude and phase x, y, z components were selected for analysis. The aortic centreline was automatically set, as was a threshold for the aortic volumes, which were manually adjusted to fit the centreline at the point of the highest flow through the aorta in systole. The plane for the analysis was set at the level of the mid-AAo (i.e., the same level of surgical biopsy). Wall shear stress on the aortic wall and flow streamlines, velocity, and flow displacement in the aorta were measured. The values of WSS were derived as 90 vectors on the analysis plane and then combined into six regions, corresponding to the six surgical segments. 

### 2.7. Statistical Analysis

Statistical analysis (R, Vienna, Austria) was carried out to compare the level of miRNAs, proteins, elastin (% and thickness), and WSS between the dilated vs. the non-dilated segments of the aorta. Non-parametric paired analysis was carried out (Wilcoxon signed-rank test). Data were reported as medians and interquartile ranges (IQRs) or proportions, as appropriate. Where the associations between variables were assessed, linear regression analysis was performed. A *p*-value < 0.05 indicated statistical significance. 

## 3. Results

### 3.1. Patient Population

The baseline characteristics of all the recruited patients (n = 15) are reported in Table 1. The patients’ age was 58 (56, 67) years (range: 30–75 years), 67% were male, and they had a BSA of 1.92 (1.85, 2.18) m^2^. The patients predominantly had anterior-right AAo dilation (80 vs. 20% overall dilated). The patients had varying degrees of aortic regurgitation (47% moderate/severe) and aortic stenosis (73% moderate/severe). Molecular data were collected in all patients, of which n = 8 were collected for full molecular characterisation. Haemodynamic data were available in n = 6 patients aged 58 (56, 66) years, 50% of whom were male, with a BSA of 1.92 (1.70, 2.13) m^2^ (Table 1). 

### 3.2. Molecular Analysis

In the NGS analysis (n = 8), a paired comparison between AR and PL segments revealed 92 differentially expressed miRNAs based on a *p*-value < 0.05 (see Appendix A), of which 11 with an FDR < 0.1 were chosen as targets for qPCR-based validation (Table 2). Highly sensitive quantification by miRCURY LNA assays in the whole cohort (n = 15) were performed to confirm differentially expressed miRNA candidates. Table 3 reports the results for the detectable miRNAs from the analysis. A paired comparison between the AR vs. the PL and the A + AR vs. the P + PL segments highlighted five differentially expressed miRNAs (miR-128-3p, miR-210-3p, miR-150-5p, miR-199b-5p, and miR-21-5p, see Table 3). Additionally, the results for the overall-dilated patient shown in Table 3 indicate no miRNA difference from the segmental comparison, compared to the one-side-dilated patients. 

Proteomic analysis in the same cohort (n = 8) revealed 579 significantly dysregulated proteins based on a *p*-value < 0.05 in the AR vs. PL segment comparison (see Appendix A). 

### 3.3. Bioinformatic Analysis

Bioinformatic analysis linking the above five miRNAs (validated by miRCURY LNA assays) and significantly dysregulated proteins revealed a potential network link between three miRNAs (miR-128-3p, miR-150-5p, miR-199b-5p) and twelve proteins (Figure 2). Gene ontology and relevant protein pathways were identified for the eight most differentially expressed proteins, as shown on the volcano plot (Figure 3; Table 4 and Table 5), based on −0.5 > log_2_FC > 0.5 and −log10(*p*-value) > 1.3. 

### 3.4. Histomorphometric Analysis

Histomorphometric analysis (same cohort, n = 8) showed that the dilated aortic segments (i.e., AR) had decreased elastic fibres [average % area of elastic fibres: AR = 45.21 (39.71, 47.64) vs. PL = 48.74 (45.22, 52.64), *p* = 0.039] and a thinner elastic layer [AR = 24.71 (20.26, 28.69) vs. PL = 29.21 (22.35, 33.36) um, *p* = 0.016] (Figure 4).

### 3.5. Haemodynamic Analysis

Where available (n = 6), a visual assessment of WSS distribution, velocity streamlines, and flow-velocity distribution (Figure 5) revealed the following: (i) an increased WSS and velocity at the anterior-right side in patients with anterior-right dilation, with helical right-handed flow and eccentric flow velocity towards the anterior-right wall; and (ii) a further increased WSS on the AAo wall in patients with overall dilation, helical aortic flow, and a higher flow velocity.

Interestingly, a correlation between increased WSS and decreased miRNA expression was found for miR-128-3p and miR-210-3p (Figure 6), with these miRNAs identified as being downregulated on the side of dilation (Table 3, see log_2_FC). In patients with both histomorphometric and haemodynamic data (n = 4), the dilated segments (i.e., those with less elastic fibres and a thinner layer) also exhibited double WSS values on average, compared to the non-dilated segments [AR vs. PL: 1523.43 (1141.55, 1934.00) vs. 786.73 (514.25, 1005.93) mPa s].

## 4. Discussion

In order to study the complex nature of BAV aortopathy, a segmental approach can be beneficial to uncover detailed local information about changes in the aortic wall. We conducted a prospective study, collecting tissue biopsies for (a) NGS and miRCURY LNA PCR for the identification and validation of miRNA markers, (b) proteomic analysis, (c) bioinformatic network analysis, (d) histomorphometric analysis of elastic fibres, and (e) preliminary 4D flow CMR data for haemodynamic assessment. The NGS and miRCURY LNA PCR validation showed that five miRNAs had significant differences between the anterior-right and posterior-left sides of the aorta, i.e., miR-128-3p, miR-210-3p, miR-150-5p, miR-199b-5p, and miR-21-5p. Our bioinformatic network analysis showed that, amongst those identified by proteomics, eight gene targets, i.e., MGAT1, PTGER3, STX5, PAPPA, SULF1, EIF5B, RASSF2, and PAN3, are linked to miR-128-3p, miR-150-5p, and miR-199b-5p. Based on a strict cut-off of ±0.5 log_2_FC (also considering the FDR), two miRNAs were significantly dysregulated, i.e., miR-128-3p (downregulated) and miR-150-5p (upregulated), while miR-210-3p (downregulated) and miR-199b-5p (upregulated) could be considered borderline.

MiR-128-3p has been characterised in the literature as a novel regulator of vascular smooth-muscle-cell (SMC) phenotypic switches, regulated by epigenetic modifications under stress [13]. While SMCs have been abundantly studied in the aneurysmal process, the role of miR-128-3 on SMCs should be further exploited for therapeutics. Based on our network analysis, miR-128-3p is linked to PTGER3 gene expression, which in turn is associated with prostanoid ligand receptors (*p* = 0.005), eicosanoid ligand-binding receptors (*p* = 0.008), the relationship between inflammation, COX-2 and EGFR (*p* = 0.013), and prostaglandin synthesis and regulation (*p* = 0.024). This shows its involvement in arachidonic acid metabolic pathways, activated by inflammation, as previously reported in aortic aneurysm formation in Marfan syndrome [14], which is in turn suggested to share similarities with BAV aortopathy [15]. Additionally, miR-150-5p is referred to in the literature as being involved in the regulation of genes with a role in eicosanoid synthesis, such as PTGIS and the MMP/TIMP pathway, which are implicated in the pathogenesis of abdominal aortic aneurysm (AAA) [16]. There is also speculation that miR-21 regulates the PTGS2/PGE2 pathway in the AAA context [16]. All this may imply an interconnected pathway of the above miRNAs and prostaglandin biosynthesis in the aneurysmal process that should be further studied. Furthermore, miR-150-5p and miR-21-5p have been previously mentioned as both plasma and tissue biomarkers of AAA [17], and the role of miR-21-5p in BAV aortopathy has been well-established in the literature [18,19,20,21].

Based on a study of thoracic aneurysmatic aortopathy in Marfan syndrome, miR-199b-5p was significantly dysregulated and associated with extracellular matrix (ECM)–receptor interaction and the Hippo signalling pathway [22]. Based on our network analysis, miR-199b-5p might be associated with Hippo signalling through RASSF2 (*p* = 0.015) [23]. The Hippo signalling pathway is involved in ECM production and degradation, as well as in the growth, death, and migration of vascular SMCs and endothelial cells, which contribute to vascular remodelling in cardiovascular diseases, such as aortic aneurysms, pulmonary hypertension, and atherosclerosis, and in cardiovascular processes, such as restenosis and angiogenesis [24]. Additionally, our network analysis showed miR-199b-5p was associated with SULF1, which has in turn been associated with vascular-endothelial growth-factor (VEGF) production and receptor-signalling pathways (*p* = 0.017 and *p* = 0.043, respectively) [25,26]. VEGF has a central role in promoting angiogenesis as it contributes to the proliferation, survival, and migration of endothelial cells and their progenitors, whilst also mediating the migration of circulating inflammatory monocytes and tissue macrophage precursors to target organs [27]. Additionally, pathway cross-talking is supported by the literature (e.g., TNF-a and RASSF–Hippo signalling, and TGF-β1 and VEGF, together with the Notch signalling pathways involved in the regulation of vascular morphogenesis). Additionally, BAV aortopathy may reflect the pathophysiological implications of BAV disease, thereby involving multiple integrated cell-signalling pathways [23,28]. These mechanisms should be further examined through in vitro cell-culturing studies. Additionally, because flow-sensitive miRNAs, known as mechanomiRs, can regulate endothelial gene expression and, consequently, endothelial dysfunction [29], the above network involving VEGF should be further examined through the relationship of miRNAs influenced by blood-flow changes.

Whether the pathogenesis of BAV aortopathy is related to genetic or haemodynamic factors is still a cause for debate [30]. A “genetic theory” supports that the brittleness of the aortic wall is caused by the developmental abnormalities of the wall and the aortic valve, due to heritable gene mutations [31,32]. In contrast, a “haemodynamic theory” focuses on abnormal wall shear stress (WSS) progressively acting on the aortic wall, with emerging evidence showing that even a ‘clinically normal’ BAV is associated with abnormal flow patterns and asymmetrically increased WSS in the proximal aorta [31,33]. Changes in WSS in BAV have been studied using CMR [34,35,36]. Our results confirmed that BAV patients had right-handed helical and more turbulent, eccentric blood flow (i.e., flow displacement from the centre towards the side of the aortic wall with increased velocity). These results concur with the literature, as BAV aortas, compared to TAV, show increased helicity in the mid-AAo and increased flow angle, associated with a rotating jet along the aortic wall that eventually leads to dilation [37]. It has also been reported that right-handed helical flow was the most common flow abnormality in BAV, associated with large AAo diameters and a high rotational flow and systolic flow angle, irrespective of valve morphotype [37]. Furthermore, BAV cusp-fusion morphology is also reported in the literature as a factor altering flow patterns, WSS, and the expression of aortopathy [38]. This could not be examined in this study due to the small available sample, but it should be further examined in larger cohort studies considering segmental alterations. 

One study examined miRNA differences between the convexity and concavity in mildly aortic-dilated BAV patients, TAV patients, and non-dilated heart transplant donors [10]. Here, aortic RNA was analysed through PCR arrays, profiling the expression of 84 microRNAs involved in cardiovascular disease, target prediction, and pathway-enrichment analysis. The study showed that miRNA dysregulation, potentially affecting mechanotransduction pathways, exhibits a higher prevalence in BAV convexity, with reference to TGF-β1, Hippo, and PI3K/Akt/FoxO. BAV convexity vs. BAV concavity showed, among others, a significant upregulation of hsa-miR-150-5p and hsa-miR-21-5p, and a significant downregulation of hsa-miR-199a-5p, as well as a significant involvement in Hippo signalling pathways, all in agreement with our results. However, the direction of miRNA dysregulation (i.e., either upregulation or downregulation) can be controversial in the literature, as in the case of miR-21 [39]. Additionally, a pathway-enrichment analysis from an miRNome-profiling study in BAV aortopathy identified a link with the Hippo signalling pathway, ErbB signalling, TGF-beta signalling, and focal adhesion, verifying the involvement of these pathways in the disease and their link with miRNA dysregulation [40].

In our population, the highest miRNA, WSS, and elastin dysregulation (decreased levels of elastic fibres and a thinner layer) was observed at the dilated anterior-right side of the aortic wall, suggesting a correspondence between molecular and haemodynamic changes. Whether these miRNAs are directly involved in the aortic wall pathogenesis of BAV disease, or whether they are a consequence of the increased aortic shear stress generated by the anomalous aortic flow caused by BAV, should be studied in larger cohorts. A key study by Guzzardi et al. [8] reported the relationship between altered WSS and regional aortic tissue remodelling (i.e., elastic fibre quantification, fibre thickness, and fibre distance, and MMP and TGF-β concentration) in BAV aortopathy in regions of the AAo (adjacent vs. distal to the right pulmonary artery) in a similarly sized group (n = 20). The presence of thin, fragmented elastic fibres, reduced fibrillin-1 content, decreased type I and III collagen, and MMP/TIMP ratio dysregulation have been well-studied in the literature, suggesting an inherent developmental defect underlying the dysregulation of the ECM in the aortic medial layer as a main cause of aortic dilatation [30,41]. Additionally, a recent study [9] correlated SMC loss, elastic degradation, and aortic wall strength to WSS, showing an increased dysregulation on the outer curve of the AAo. However, it has not yet been explained whether the WSS-mediated ECM dysregulation is a consequence of the dilatation itself, rather than a determinant or risk factor. Our detailed, segment-by-segment analysis advances some of these observations within an overall framework of exploring the relationship between altered haemodynamics and aortic tissue remodelling. More studies, such as the one by Zhang et al. [42] which shows that the inhibition of miRNA-29 enhances elastin levels in cells, ought to be carried out to elucidate the interaction between mechanomiRs and elastin and demonstrate that miRNAs can be used in therapeutics and the discovery of novel pharmacological targets [43].

Further studies are needed to prove that vascular remodelling could be a biomarker in clinical imaging and miRNAs could be the effectors of flow-mediated changes. Here, we have proposed a specific segmental comparison of A + AR vs. P + PL (segments with the most vs. the least molecular dysregulation, as derived from our analyses) and have shown segmental correspondence between miRNAs and WSS. We have also provided additional information on proteins and elastin changes, suggesting miRNAs as mechanistic mediators of the vascular changes induced by the altered flow in BAV. However, we do not know whether these miRNA changes have a transcriptional or post-transcriptional origin. It would therefore be interesting to understand this through a follow-up study, as it could provide information on cellular processes involved in the dysregulation and, ultimately, on BAV aortopathy and its genetic vs. WSS drivers. 

From a methodological standpoint, we note extending observations from AR vs. PL segments to A + AR vs. P + PL was based on (a) ensuring to cover the full spectrum of the dilated area, and (b) accounting for potential human error in the process of segmentation in the surgical theatre, although the latter was carefully guided by and discussed with surgeons. The study, due to the prospective design, has the disadvantage of not all the recruited patients having a full molecular and haemodynamic dataset. The lack of these data was mainly caused either by issues in collecting the biopsies at the time of surgery or patients not consenting to the additional CMR scan. 

## 5. Conclusions

This study merged segmental molecular and histological data, as well as clinical imaging, shedding light on local changes across the aortic wall circumference in BAV patients. MiRNA dysregulation linked to inflammatory pathways, changes in aortic wall remodelling, and WSS were concomitantly observed on the anterior-right wall in BAV patients, emphasizing the requirement to further study regional changes in the aorta.

## Figures and Tables

**Figure 1 cells-11-03721-f001:**
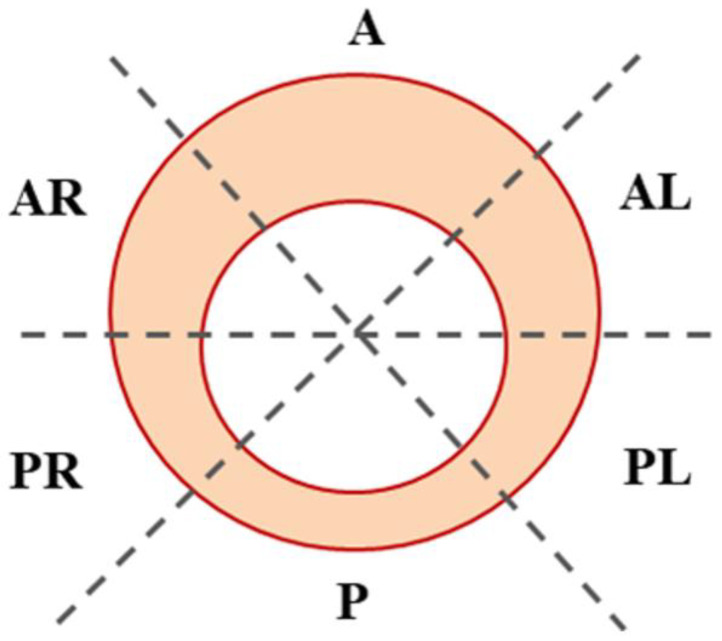
Map of the ascending aorta segmentation. Six segments were cut for each aortic sample, i.e., anterior (A), anterior-right (AR), posterior-right (PR), posterior (P), posterior-left (PL), and anterior-left (AL), and preserved for analyses.

**Figure 2 cells-11-03721-f002:**
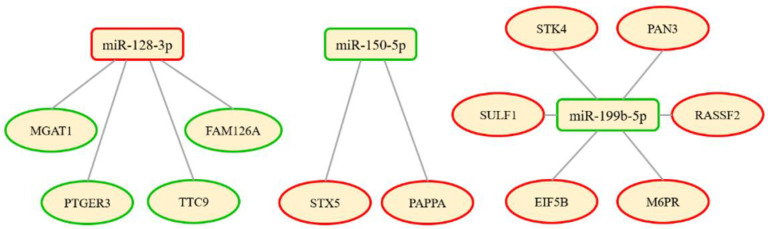
MiRNA and protein network link. The networks indicate predicted interactions between validated miRNAs and the significantly dysregulated proteins. Green = upregulation; red = downregulation.

**Figure 3 cells-11-03721-f003:**
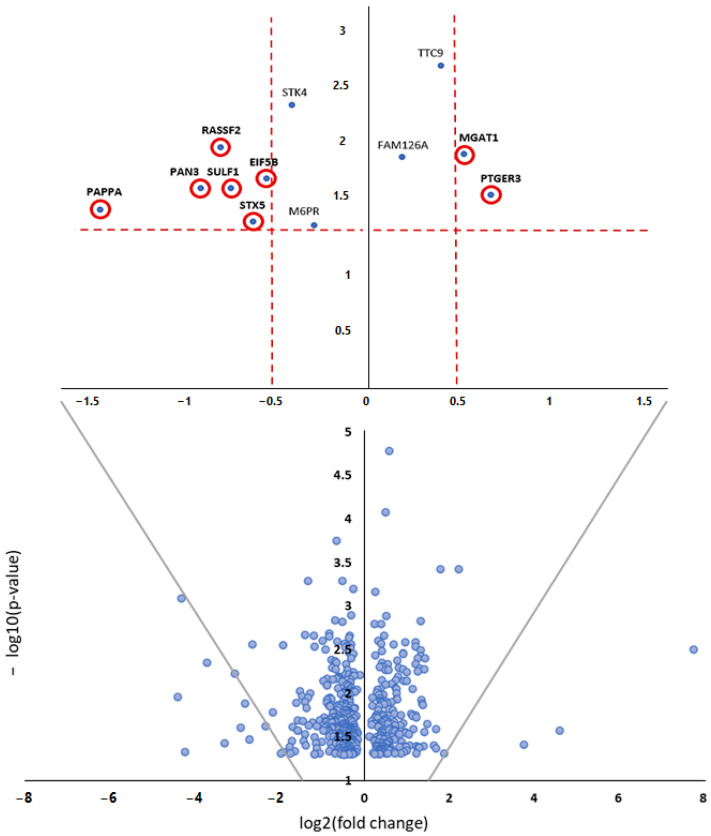
Volcano plot for the selection of the differentially expressed proteins. The 579 proteins with a *p*-value < 0.05 identified from the proteomic analysis (see Appendix A) are displayed. Extracted and plotted alone on top of the plot are the significantly dysregulated proteins (highlighted in red) as determined by cut-off values of −0.5 > log2FC > 0.5 and −log10(*p*-value) > 1.3. Proteins with log2FC < 0 are downregulated in dilated vs. non-dilated segments, while proteins with log_2_FC > 0 are upregulated.

**Figure 4 cells-11-03721-f004:**
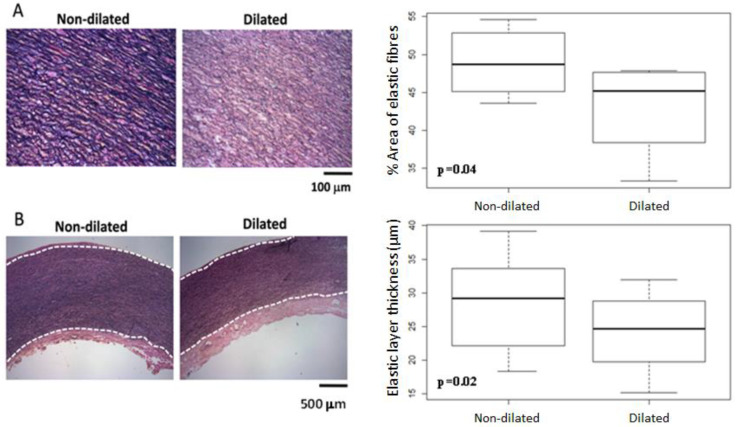
Quantitative histomorphometry of the elastic component. (**A**) Representative images (200× magnification) for non-dilated and dilated segments, and differences (% area) of elastic fibres showing decreased elastic fibres in dilated segments (*p* = 0.04); (**B**): example of measurements of elastic layer (40× magnification) and differences in thickness showing a thinner layer in dilated segments (*p* = 0.02).

**Figure 5 cells-11-03721-f005:**
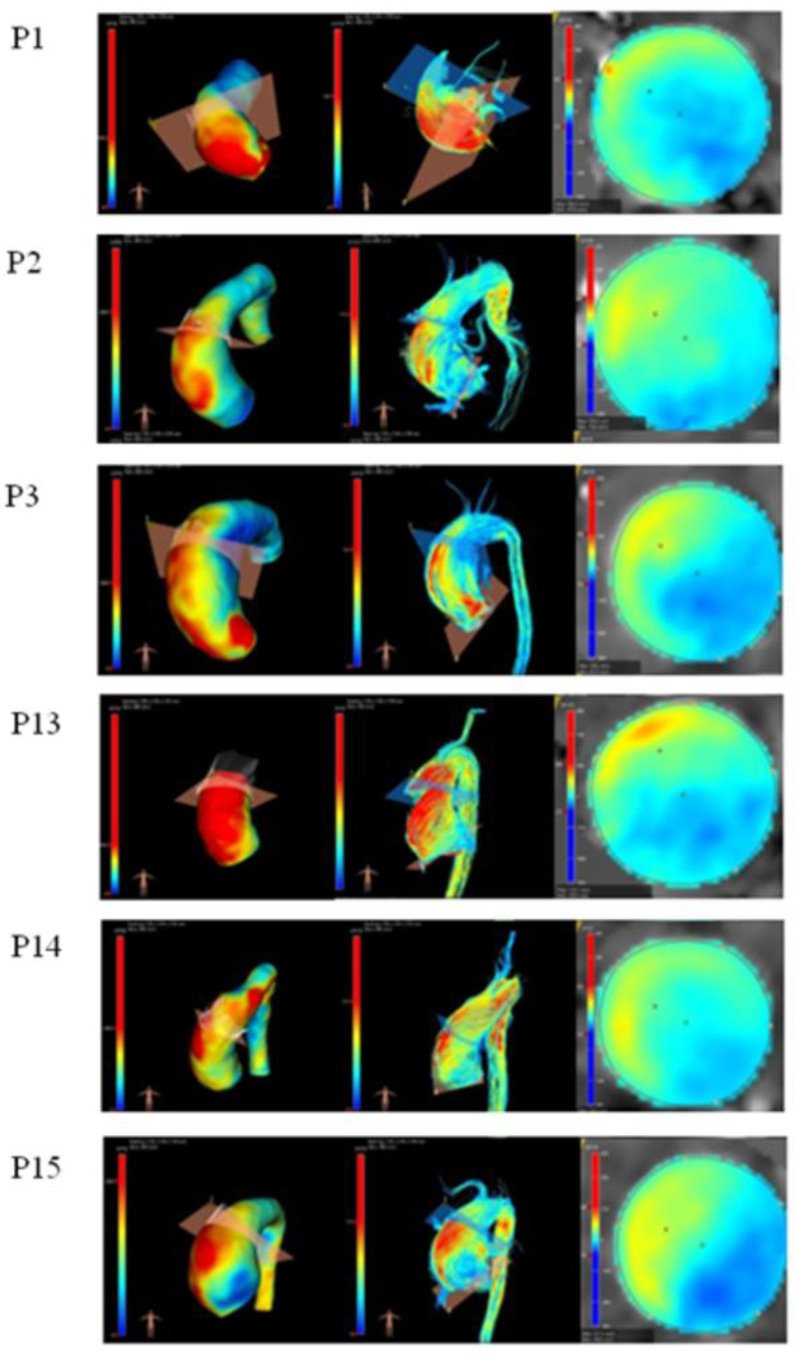
4D flow analysis in patients with both molecular and haemodynamic data. First column: wall shear stress (WSS, range = 0–2000 mPAŸs); second column: velocity streamlines (range = −122–262 cm/s); third column: flow displacement.

**Figure 6 cells-11-03721-f006:**
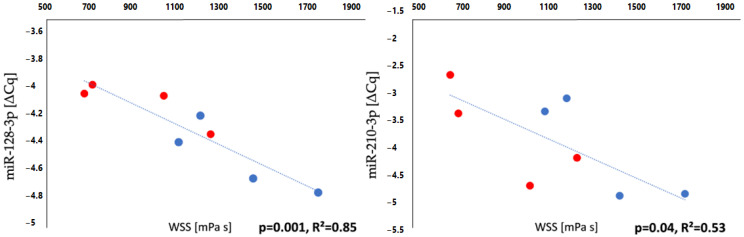
Correlation between WSS and miRNA. Relationship between WSS and miRNA expression in n = 4 patients with anterior-right dilation. Red = average of posterior segments; blue = average of anterior segments. Both graphs show less miRNA expression and higher WSS in the dilated segments.

**Table 1 cells-11-03721-t001:** Patient population characteristics.

Patient	Sex	Age at the Date of Study (yrs)	BSA (m^2^)	Type of Surgery	Type of Dilation	Degree of Dilation (echo mm)	Valve Phenotype	Aortic Regurgitation Severity	Aortic Stenosis Severity	Presence of Calcification	NGS, Proteomics, Histology	miRCURY LNA Assay	Haemodynamic Data
P2	F	69	1.86	AVR, AAo, AF	Anterior-Right	SoV:42 AAo:49	RL	Moderate	None	None	✓	✓	✓
P3	M	68	1.98	AVR, AAo	Anterior-Right	SoV:38 AAo:42	RNC	None	Severe	Yes	✓	✓	✓
P6	M	58	2.18	AVR, AAo	Anterior-Right	SoV:33 AAo:40	RL	None	Moderate	Yes	✓	✓	
P7	F	66	1.84	AVR, Arch	Anterior-Right	SoV:23 AAo:44	RNC	Mild	Mild	None	✓	✓	
P8	M	59	1.88	AVR, AAo/Root	Anterior-Right	SoV:44 AAo:41	RL	None	Severe	Yes	✓	✓	
P9	M	65	2.35	AVR, AAo, CAN	Anterior-Right	SoV:44 AAo:49	RL	Moderate	None	None	✓	✓	
P14	F	57	1.55	Ross	Anterior-Right	SoV:31 AAo:36	RL	None	Severe	None	✓	✓	✓
P15	F	30	1.65	Ozaki, AAo	Anterior-Right	SoV:37 AAo:37	RL	Moderate	Mild	None	✓	✓	✓
P1	M	55	2.19	AVR, AAo, AF, LAA	Overall	SoV:38 AAo:46	RNC	Moderate	Moderate	None		✓	✓
P4	F	32	1.9	Ross	Anterior-Right	SoV:39 AAo:38	RNC	Severe	Severe	Yes		✓	
P5	M	34	1.92	AVR, AAo	Anterior-Right	SoV:34 AAo:45	RNC	Mild	Severe	Yes		✓	
P10	M	70	1.78	AVR, AAo	Overall	SoV:40 AAo:46	RL	Mild	Severe	Yes		✓	
P11	M	75	2.18	Root	Anterior-Right	SoV:42 AAo:51	RNC	Moderate	Moderate	Yes		✓	
P12	M	56	1.99	AVR, AAo	Anterior-Right	SoV:36 AAo:45	N/A	Moderate	Severe	Yes		✓	
P13	M	58	2.19	Ozaki, AAo	Overall	SoV:46 AAo:53	RNC	None	Severe	Yes		✓	✓

Baseline characteristics for patients (P), including data collection. AAo = ascending aorta; AF = atrial fibrillation; AVR = aortic valve replacement; CAN = coronary artery bypass; LAA = left atrial appendage; N/A = non-applicable; RL = right-left; RNC = right-non-coronary; SoV = sinus of Valsalva.

**Table 2 cells-11-03721-t002:** Next-generation sequencing chosen targets for miRCURY LNA assay validation.

MiRNA Name	Dilated vs. Non-Dilated—FC	Dilated vs. Non-Dilated—Log_2_FC	Dilated vs. Non-Dilated—*p*-Value	Dilated vs. Non-Dilated—FDR *p*-Value
**hsa-miR-1247-5p**	3.5892	1.8437	<0.0001	0.0003
**hsa-miR-21-5p**	0.6667	−0.5849	<0.0001	0.0003
**hsa-miR-483-3p**	2.1387	1.0967	<0.0001	0.0024
**hsa-miR-211-5p**	0.3308	−1.5960	<0.0001	0.0095
**hsa-miR-21-3p**	0.6068	−0.7206	<0.0001	0.0095
**hsa-miR-199b-5p**	2.1820	1.1256	<0.0001	0.0095
**hsa-miR-128-3p**	0.6975	−0.5197	<0.0001	0.0095
**hsa-miR-150-5p**	1.8983	0.9247	0.0002	0.0431
**hsa-miR-488-3p**	2.7460	1.4573	0.0003	0.0615
**hsa-miR-375-3p**	2.5455	1.3480	0.0005	0.0872
**hsa-miR-210-3p**	0.6465	−0.6294	0.0006	0.0996

FC = fold change; FDR = false discovery rate.

**Table 3 cells-11-03721-t003:** miRCURY LNA validated targets from segmental analysis.

	*AR vs. PL*					*A + AR vs. P + PL*				
	Anterior-Right (N = 12)				Overall (N = 3)	Anterior-Right (N = 12)				Overall (N = 3)
*MiRNA Targets NGS*	*p*-Value	FC	Log_2_FC	FDR	*p*-Value	*p*-Value	FC	Log_2_FC	FDR	*p*-Value
*miR-1247-5p*	0.110	1.647	0.720	0.154	0.750	0.470	1.291	0.368	0.470	0.470
*miR-21-5p*	0.339	0.919	−0.121	0.396	0.250	**0.034**	0.835	−0.261	**0.060**	0.250
*miR-21-3p*	0.791	1.030	0.043	0.791	0.500	0.380	0.869	−0.202	0.444	0.500
*miR-199b-5p*	**0.012**	1.354	0.437	**0.021**	0.500	0.233	1.173	0.230	0.327	1.000
*miR-128-3p*	**0.012**	0.724	−0.466	**0.021**	0.250	**<0.001**	0.665	−0.588	**0.002**	0.750
*miR-150-5p*	**0.007**	1.736	0.796	**0.021**	0.750	**0.012**	1.418	0.504	**0.028**	1.000
*miR-210-3p*	**0.009**	0.773	−0.371	**0.021**	0.250	**<0.001**	0.722	−0.471	**0.002**	0.750

Reporting two comparisons of aortic segments (anterior-right vs. posterior-left, AR vs. PL, and a more extensive comparison combining adjacent segments, anterior + anterior-right vs. posterior + posterior-left, A + AR vs. P + PL) for one-side- and overall-dilated patients. In bold, *p*-value < 0.05 and FDR < 0.1, which were used as threshold of significance.

**Table 4 cells-11-03721-t004:** Gene ontology analysis for the eight most significantly dysregulated proteins.

Gene Target	Description	Functional Database	*p*-Value
*MGAT1*	peptidyl-asparagine modification	Biological_Process	0.0139
*PAPPA*	response to gonadotropin	Biological_Process	0.0144
*MGAT1; SULF1*	glycoprotein metabolic process	Biological_Process	0.0150
*SULF1*	vascular-endothelial growth-factor production	Biological_Process	0.0169
*PTGER3; SULF1*	muscle system process	Biological_Process	0.0176
*MGAT1*	nucleotide sugar metabolic process	Biological_Process	0.0184
*MGAT1*	amino sugar metabolic process	Biological_Process	0.0188
*SULF1*	animal organ formation	Biological_Process	0.0301
*SULF1*	nerve development	Biological_Process	0.0379
*SULF1*	vascular-endothelial growth-factor receptor-signalling pathway	Biological_Process	0.0427
*PTGER3*	phospholipase C-activating G-protein-coupled receptor-signalling pathway	Biological_Process	0.0470
*STX5*	organelle disassembly	Biological_Process	0.0470
*MGAT1*	catalytic activity, acting on a glycoprotein	Molecular_Function	0.0114
*STX5*	SNAP receptor activity	Molecular_Function	0.0149
*MGAT1*	manganese ion binding	Molecular_Function	0.0301
*EIF5B*	translation factor activity, RNA binding	Molecular_Function	0.0427
*STX5*	SNARE binding	Molecular_Function	0.0500
*STX5*	SNARE complex	Cellular_Component	0.0233

Bioinformatic tools were used to extract gene ontology analysis for the eight most differentially expressed proteins associated with the three miRNA targets, (*p*-value < 0.05).

**Table 5 cells-11-03721-t005:** Pathway analysis for the eight most significantly dysregulated proteins.

Gene Target	Description	*p*-Value
*MGAT1*	N-glycan trimming and elongation in the cis-Golgi	0.0027
*MGAT1; STX5*	transport to the Golgi and subsequent modification	0.0039
*PTGER3*	prostanoid ligand receptors	0.0048
*PTGER3*	eicosanoid ligand-binding receptors	0.0080
*PTGER3*	small ligand GPCRs	0.0102
*MGAT1; STX5*	asparagine N-linked glycosylation	0.0104
*PTGER3*	relationship between inflammation, COX-2, and EGFR	0.0133
*PAN3*	deadenylation of mRNA	0.0133
*RASSF2*	Hippo signalling pathway	0.0155
*STX5*	Cargo concentration in the ER	0.0176
*STX5*	SNARE interactions in vesicular transport	0.0181
*STX5*	intra-Golgi traffic	0.0234
*PTGER3*	prostaglandin synthesis and regulation	0.0239
*MGAT1*	N-glycan biosynthesis	0.0260
*EIF5B*	translation factors	0.0286
*PTGER3*	regulation of lipolysis in adipocytes	0.0286
*PAN3*	deadenylation-dependent mRNA decay	0.0297
*STX5*	COPII-mediated vesicle transport	0.0359
*PAN3*	RNA degradation	0.0416

Bioinformatic tools were used to extract pathway analysis for the eight most differentially expressed proteins associated with the three miRNA targets (*p*-value < 0.05).

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
