# Peer review of "A Segmental Approach from Molecular Profiling to Medical Imaging to Study Bicuspid Aortic Valve Aortopathy"

_cells, 2022, doi:10.3390/cells11233721_

Round 1

Reviewer 1 Report

1)      It is unclear, how the aortopathy was defined? Some patients had maximal aortic diameter < 40mm

2)      Very small sample sizes, 8 patients with full molecular analysis and 6 patients with imaging data

3)      This a cross-sectional study and, therefore, no causality can be stated. The same applies to the fact that only patients with BAV aortopathy were included and, therefore, the reported molecular and rheological changes might be a sequel of aortic dilatation and not vice versa.

Author Response

Reviewer 1

1)      It is unclear, how the aortopathy was defined? Some patients had maximal aortic diameter < 40mm

We thank the Reviewer for the comment and the chance to explain. The choice was based on clinical assessment. All patients had ascending aortic diameter of >40 mm, except for two patients whose ascending aorta was 36-37mm. Not only this is borderline dilated, but the surgeon assessment was to replace the aorta because of family history and an observed rapid increase in aortic dilatation over the last 2 years. Patients with severe regurgitation or stenosis have a replacement of ascending aorta at 4.5 cm, whereas asymptomatic BAV patients may replace the aorta at 5.0-5.5 cm. However, when extra risk factors such as family history are involved, and the risk of dissection is high enough, this warrants operative intervention. [2020 ACC/AHA Guideline for the Management of Patients With Valvular Heart Disease: A Report of the American College of Cardiology/American Heart Association Joint Committee on Clinical Practice Guidelines]. We would like to note that most of the patients included were replacing the aortic valve, and the aorta (due to standard clinical practice protocol) was re-measured in the surgical theatre to decide whether to replace, based also on the clinical history of the patient. 

We added to the text, (page 6, Methods, Patient Population): The decision for aortic replacement follows ACC/AHA Guidelines 2020 (12).

2)      Very small sample sizes, 8 patients with full molecular analysis and 6 patients with imaging data

We agree with the Reviewer that the sample size is small, and this is due to the complex study design to cover both molecular and imaging data, in dilated patients with bicuspid aortic valve in a single centre. Also, we were advised to send a cohort of n=5 for next generation or other profiling analyses. Nonetheless, we acknowledge this point in Limitations, and anticipating that the nature of the findings (such as the identified targets and network, and the link to imaging data) and the methodological implications (such as the segmentation and sample-storage approach) are novel and noteworthy.

3)      This a cross-sectional study and, therefore, no causality can be stated. The same applies to the fact that only patients with BAV aortopathy were included and, therefore, the reported molecular and rheological changes might be a sequel of aortic dilatation and not vice versa.

Indeed only BAV-dilated patients were included in this study, and each patient acted as their own control, i.e. the comparison has been done between dilated to non-dilated segments from the same patient, rather than comparing patients with vs. without dilation. The side of dilation was specified by the surgeon at the time of the surgery when the aorta was visualised for maximum precision. In this study we show for the first time the distribution of changes across the BAV aortic circumference, and from a methodological standpoint we propose an approach of segmentation and comparison which has been agreed and revised between basic scientists/bioengineers and clinicians to show BAV dilated vs. non-dilated segmental differences. We anticipate that the findings from this work will be used for further studies using larger cohorts, by our group and others. Whilst we did not aim to explore any causal relationship in this study, we aimed to explore the very local nature of some of these changes.  

Reviewer 2 Report

The manuscript submitted by Sophocleous et al aimed to identify regional ascending aortic changes in BAV aortopathy using a segmental approach, integrating molecular data (obtained by Next-generation RNA-sequencing and Tandem Mass Spectrometric analysis), histomorphometric analysis (elastin fragmentation and degeneration), and clinical imaging (wall shear stress) in BAV patients with aortopathy. They find the dysregulation, across the aortic wall circumference, of several miRNAs linked to inflammatory pathways, changes in aortic wall remodeling and WSS on the anterior/right wall in BAV patients, emphasizing the requirement to further study regional changes in the aorta. This study is well designed and the findings are interesting and novel, but several points should be addressed.

Major comments

1.- It should be explained why authors have selected AR vs PL for the analysis when Figure 1 shows that A and P segments have the greatest difference in dilation. Is it because patients predominantly had AR AAo dilation that authors compare AR vs PL? In addition, it should be explained why A+AR vs. P+PL was chosen for one analysis. Perhaps it could be all posterior segments vs. all anterior segments like it is doing later. Were other comparisons made to confirm that there were no differences in miRNA expression in the other segments?

2.- Who were the 8 selected patients and how were they selected? Are they part of the list of 15?

3.- If non-parametric analysis is used, mean and SD are not appropriate to give information on the variables. The statistics section should be more descriptive. For example, how the fold changes of the miRNAs were calculated, why they used logFC or log2FC, and the criteria for the FDR (FDR<1 [line 183], FDR>0.05 [line 196].

4.- Regarding the molecular analysis, it would be nice to have the list of 92 differentially expressed miRNAs in supplementary material.

5.- The criterion to select the 11 miRNAs in Table 2 is conflicting. Considering the very low p-value of the last miRNA in the list, it seems that a lot of additional miRNAs should be in the table; considering the FDR-p-value, only the first 8 miRNAs should appear (although the criteria for the FDR are not clear); taking into account Bonferroni, only the first three miRNAs stay significant. Why authors use Bonferroni correction if it is not considered? This point should be better clarified in order to continue with the workflow.

6.- The next selection of 7 from 11 miRNAs is also conflicting (Table 3). Based on what criteria? As it is stated in lines 187-188, only 4 miRNAs (miR-199b-5p, miR-128-3p, miR-150-5p and miR-210-3p) are significantly dysregulated in the AR vs PL comparison and, maybe, miR-21-5p is dysregulated in the A+AR vs. P+PL comparison (taking into account the p-value that is the only one in bold). Therefore, miR-210-3p and miR-199b-5p are not borderline. It is of interest, as well, that it was any difference in the expression of miRNAs in patients with overall dilation. The whole 3.2 paragraph should be rewritten. In addition, reconsider the remarking in bold the FC when it is not significant.

7.- The list of 579 significantly dysregulated proteins may be of interest for the research community and could be incorporated as supplementary material.

8.- Results regarding Figures 2 and 3 are unclear. Figure 2 shows the potential link-network between three miRNAs and twelve “significantly dysregulated proteins”. But, in Figure 3, four of these proteins are not significantly dysregulated (STK4, M6PR, TTC9 AND FAM126A). Maybe the volcano plot should show the 579 significantly dysregulated proteins in AR vs PL and then, the authors could highlight the 8 proteins that, in a later analysis, appear as potentially regulated by the three validated miRNAs.

9.- About the results in Figure 6:

First: “a correlation between increasing WSS and increased miRNA expression was found for miR-128-3p and miR-210-3p” is only true for anterior-right dilated patients.

Second: Are difference (d) in values with respect to the P segment absolute values? Because if not, how authors explain that miR-128 and miR-210 are upregulated in AR aortic fragment, positively correlating with WSS, when miR-128 and miR-210 were downregulated in Table 3? This graphic is confusing because miR-150 and miR-199 have the same trend than miR-128 and miR-210 but the former were upregulated in Table 3.

Third: How many patients are represented in figure 6A? They should be the n=4 with haemodynamic data (WSS). But then, this information should be in the A section of Figure 6 caption.

Fourth: In what units are the miRNAs expressed in Figure 6B? The graph represents a negative correlation but R2 values are positive.

10.- Authors should consider to discuss this paper: Int J Mol Sci. 2017 Nov; 18(11): 2498. doi: 10.3390/ijms18112498, the first in-depth screening of the whole miRNome in TAA specimens and identified specific dysregulated miRNAs in BAV patients.

 Minor comments

11.- References 4 (line 55), 7 (line 58), 14-16 (line 273), and 20 (line 284) are not suitable. Also, eliminate reference 46.

12.- Display BSA in line 105. However, this variable is not used in the whole study so, maybe it should be removed.

13.- Maybe ceramic beads are 2.8 mm? (line 114).

14.- Remove the first table with 9 patients in Table 1.

15.- Also in Table 1, display SoV, N/A. There are no volunteers (V) in Table 1.

16.- In Table 4, what does the p-values correspond to and what is the p<0.05 in the footnote? In addition, remove “as shown in Figure 1”, which represents the map of the ascending aorta segmentation.

17.- Change TGER3 by PTGER3 in Table 5. Separate the footnote from the main text (line 220).

18.- In Figure 4, check the size of the bar in A, with 200X magnification. It has to be less than in B.

Author Response

Reviewer 2

The manuscript submitted by Sophocleous et al aimed to identify regional ascending aortic changes in BAV aortopathy using a segmental approach, integrating molecular data (obtained by Next-generation RNA-sequencing and Tandem Mass Spectrometric analysis), histomorphometric analysis (elastin fragmentation and degeneration), and clinical imaging (wall shear stress) in BAV patients with aortopathy. They find the dysregulation, across the aortic wall circumference, of several miRNAs linked to inflammatory pathways, changes in aortic wall remodeling and WSS on the anterior/right wall in BAV patients, emphasizing the requirement to further study regional changes in the aorta. This study is well designed and the findings are interesting and novel, but several points should be addressed.

We thank the Reviewer for the kind comments complimenting our work, and happy to reply to any comments to improve the manuscript.

Major comments

1.- It should be explained why authors have selected AR vs PL for the analysis when Figure 1 shows that A and P segments have the greatest difference in dilation. Is it because patients predominantly had AR AAo dilation that authors compare AR vs PL?

In addition, it should be explained why A+AR vs. P+PL was chosen for one analysis. Perhaps it could be all posterior segments vs. all anterior segments like it is doing later.

Were other comparisons made to confirm that there were no differences in miRNA expression in the other segments?

The most common type of bicuspid aortic valve dilation, as defined by clinicians, is the asymmetric anterior-right dilation, i.e., anterior to right wall. To take into consideration the ‘human error’ during the aortic segmentation in surgical theatre, and to fully cover the spectrum of the dilated area, we thought not only to focus on the AR segment, but to also include the A segment together with the AR by averaging the findings. Since A and AR were defined as the areas of dilation by the surgeons, P and PL were used for comparison, as the exact opposite segments to the dilated area. To simplify this already complex analysis, we limited our published results to these comparisons. Figure 6 (A) shows the average segmental distribution of the individual miRNAs and WSS levels in the population, proving our point.

We added to the text, (Methods, Molecular Analyses): As per clinicians, the anterior-right wall is dilated in the case of asymmetric BAV dilation, thus, to explore the full spectrum of the dilated area, also considering the human error in the process of segmentation, A and AR segments were defined as dilated. Therefore, comparisons were carried out for the exact opposed segments (AR vs. PL) as well as combining adjacent segments (A+AR vs. P+PL).

2.- Who were the 8 selected patients and how were they selected? Are they part of the list of 15?

The three last columns of Table 1, named as “NGS, proteomics, histology”, “miRCURY LNA assay”, and “Haemodynamic Data”, show what data are available for each patient. The 8 patients who were selected for the main core of analyses are P 2, 3, 6, 7, 8, 9, 14, 15. These patients were selected based on: the presence of asymmetric (AR) dilation, equal number of males and females to avoid gender bias, no severe mix valve disease, no severe calcification.

Correctly the Reviewer mentions that the 8 patients are part of the 15 patient-cohort used to validate the differentially expressed targets identified from NGS, using a different technique (i.e., MIRCURY LNA assay).

Addition to manuscript (Methods, Molecular Analyses): …was performed for n=8 patients (Table 1: P2, 3, 6, 7, 8, 9, 14, and 15), two opposed segments per patient i.e., AR vs. PL. These patients were selected based on the presence of asymmetric (AR) dilation, equal number of males and females to avoid gender bias, no severe mix valve disease, and no severe calcification. The RNA extract was sent to Qiagen for NGS analysis and selected markers were then validated by automated miRCURY LNA miRNA PCR assay (Qiagen) (11) by expanding the existed cohort to n=15 patients, on all 6 aortic segments per patient. Additional patients are also listed in Table 1.

We understand that Table 1 is too big to be visualised in the manuscript thus we added a screenshot for the Reviewers in the text, and we will upload the full table separately in the submission process.

3.- If non-parametric analysis is used, mean and SD are not appropriate to give information on the variables. The statistics section should be more descriptive. For example, how the fold changes of the miRNAs were calculated, why they used logFC or log2FC, and the criteria for the FDR (FDR<1 [line 183], FDR>0.05 [line 196].

We thank the Reviewer and the means and standard deviations have been replaced with medians and interquartile ranges to describe non-parametric data.

Methods, Statistical Analysis: Data are reported as medians and interquartile ranges (IQR), or proportions, as appropriate.

The statistical description of the non-parametrical data as median (IQR) have been changed in the Results for age and BSA, for the elastic component, and for the WSS displayed results were appropriate.

We thank the Reviewer and aim to clarify the above.

The fold changes were calculated as:

ΔCq = Cq(averaged normalisers) – Cq(miRNA of interest)

ΔΔCq = ΔCq(dilated segment) – ΔCq(non-dilated segment)

If ΔΔCq is positive, then FC = 2 ^ ΔΔCq

If ΔΔCq is negative, then FC = - [2 ^ (-ΔΔCq)]

The above calculations have been added in the supplementary material (miRNA data analysis).

We thank the Reviewer for identifying two mistakes in the manuscript. We have now corrected Table 2 “Dilated vs. Non-dilated – LogFC” to “Dilated vs. Non-dilated – Log2FC”; and Table 3 description “FDR>0.05” to “FDR<1”.

The Log2FC is shown on the table to provide the direction of the change (up or down-regulation. The FDR cutoff used for the choice of miRNA targets from the NGS was 1, to allow us to validate more targets with very significant p-value who were involved in aortopathy in the literature. 

We have updated the text, (Results, Molecular Analysis) to: In the NGS analysis (n=8), paired comparison between AR and PL segments revealed 92 differentially expressed miRNAs based on p-value<0.05, of which 11 with FDR<1 were chosen as targets for qPCR-based validation (Table 2). Highly sensitive quantification by miRCURY LNA assays in the whole cohort (n=15) performed to confirm differentially expressed miRNA candidates. Paired comparison of AR vs PL and A+AR vs P+PL segments highlighted 5 differentially expressed miRNAs (miR-128-3p, miR-210-3p, miR-150-5p, miR-199b-5p, and miR-21-5p, see Table 3). As further indicated by log2FC and FDRs, miR-128-3p and miR-150-5p were significantly dysregulated, while miR-210-3p and miR-199b-5p were borderline.

4.- Regarding the molecular analysis, it would be nice to have the list of 92 differentially expressed miRNAs in supplementary material.

The list of the 92 differentially expressed miRNAs has now been added to the supplementary material as requested by the Reviewer (attached as an excel file, due to its size, to be added to the Appendix A).

5.- The criterion to select the 11 miRNAs in Table 2 is conflicting. Considering the very low p-value of the last miRNA in the list, it seems that a lot of additional miRNAs should be in the table; considering the FDR-p-value, only the first 8 miRNAs should appear (although the criteria for the FDR are not clear); taking into account Bonferroni, only the first three miRNAs stay significant. Why authors use Bonferroni correction if it is not considered? This point should be better clarified in order to continue with the workflow.

Since the p-value was significant for many miRNAs, the choice of targets to further validate was based on FDR<1. Bonferroni was added on Table 2 for completeness, but since this was not a multiple comparison, the choice of targets was not based on Bonferroni. Following Reviewer’s advice for clarification of the results we removed Bonferroni from Table 2.

6.- The next selection of 7 from 11 miRNAs is also conflicting (Table 3). Based on what criteria? As it is stated in lines 187-188, only 4 miRNAs (miR-199b-5p, miR-128-3p, miR-150-5p and miR-210-3p) are significantly dysregulated in the AR vs PL comparison and, maybe, miR-21-5p is dysregulated in the A+AR vs. P+PL comparison (taking into account the p-value that is the only one in bold). Therefore, miR-210-3p and miR-199b-5p are not borderline. It is of interest, as well, that it was any difference in the expression of miRNAs in patients with overall dilation. The whole 3.2 paragraph should be rewritten. In addition, reconsider the remarking in bold the FC when it is not significant.

The miRCURY LNA assay indeed validated 5 out of the 11 targets as per Table 3. MiR-1247-5p was shown on the table due to the fact it was the most significantly dysregulated target from NGS, and from miRCURY LNA assay showed high fold change. The miR-21-3p was shown for a complete picture of miR-21, along with miR-21-5p. However, we understand the point of the Reviewer, and to simplify the results for the reader we now show only the significant miRNAs, we have remarked in bold when not significant, and re-wrote the paragraph as per comment 3 reply:

We have updated the text, (Results, Molecular Analysis) to: In the NGS analysis (n=8), paired comparison between AR and PL segments revealed 92 differentially expressed miRNAs based on p-value<0.05, of which 11 with FDR<1 were chosen as targets for qPCR-based validation (Table 2). Highly sensitive quantification by miRCURY LNA assays in the whole cohort (n=15) performed to confirm differentially expressed miRNA candidates. Paired comparison of AR vs PL and A+AR vs P+PL segments highlighted 5 differentially expressed miRNAs (miR-128-3p, miR-210-3p, miR-150-5p, miR-199b-5p, and miR-21-5p, see Table 3). As further indicated by log2FC and FDRs, miR-128-3p and miR-150-5p were significantly dysregulated, while miR-210-3p and miR-199b-5p were borderline.

7.- The list of 579 significantly dysregulated proteins may be of interest for the research community and could be incorporated as supplementary material.

The list of the 579 significantly dysregulated proteins has now been added to the supplementary material as requested by the Reviewer (attached as an excel file, due to its size, to be added to the Appendix A).

8.- Results regarding Figures 2 and 3 are unclear. Figure 2 shows the potential link-network between three miRNAs and twelve “significantly dysregulated proteins”. But, in Figure 3, four of these proteins are not significantly dysregulated (STK4, M6PR, TTC9 AND FAM126A). Maybe the volcano plot should show the 579 significantly dysregulated proteins in AR vs PL and then, the authors could highlight the 8 proteins that, in a later analysis, appear as potentially regulated by the three validated miRNAs.

We have already tried before what is being suggested, indeed agreeing with the Reviewer, but it is not visually possible for this particular case to show all the 579 and spot the 8 differentially expressed having a miRNA link network. In figure 2 we show the results from the link-network analysis between the differentially expressed miRNAs and the whole spectrum of dysregulated proteins in our population, ending up with links between 3 miRNAs and 12 proteins. Then in figure 3, we show that only 8 out of the 12 proteins are differentially expressed.

Nonetheless, we included the volcano plot of the 579 protein-list as requested in the supplementary material along with the full list of proteins. 

9.- About the results in Figure 6:

First: “a correlation between increasing WSS and increased miRNA expression was found for miR-128-3p and miR-210-3p” is only true for anterior-right dilated patients.

Second: Are difference (d) in values with respect to the P segment absolute values? Because if not, how authors explain that miR-128 and miR-210 are upregulated in AR aortic fragment, positively correlating with WSS, when miR-128 and miR-210 were downregulated in Table 3? This graphic is confusing because miR-150 and miR-199 have the same trend than miR-128 and miR-210 but the former were upregulated in Table 3.

Third: How many patients are represented in figure 6A? They should be the n=4 with haemodynamic data (WSS). But then, this information should be in the A section of Figure 6 caption.

Fourth: In what units are the miRNAs expressed in Figure 6B? The graph represents a negative correlation but R2 values are positive.

We would like to apologise for any confusion. It is true that the figure 6 (B) shows only anterior-right dilated patients (n=4). The miRNA levels are expressed as ΔCq and the calculation is: ΔCq = Cq(average normalisers) – Cq(miRNA of interest). This means that a miRNA with a more negative ΔCq will be downregulated with respect to a less negative one. As per figure legend, blue dots show anterior (dilated) segments, whereas red dots show posterior (non-dilated) segments.

We have now added on the miRNA axis “[ΔCq]” to explain what is shown.

We have not indicated the graph direction as we don’t report the R but the R2 value which is always positive.

We understand the confusion created around figure 6 A and B, and since the text in the results section is on the part B, we decided to remove the part A as it does not add to the manuscript.

We added in the text:  Interestingly, a correlation between increasing WSS and decreased miRNA expression was found for miR-128-3p and miR-210-3p (Figure 6), while these miRNAs identified as downregulated in the side of dilation (Table 3, see log2FC).

Also, Figure 6 has been updated as: Correlation between WSS and miRNA

Figure 6 legend: Relationship between WSS and miRNA expression in n=4 patients with anterior-right dilation. Red = average of posterior segments, blue = average of anterior segments. Both graphs show less miRNA expression and higher WSS in the dilated segments.

10.- Authors should consider to discuss this paper: Int J Mol Sci. 2017 Nov; 18(11): 2498. doi: 10.3390/ijms18112498, the first in-depth screening of the whole miRNome in TAA specimens and identified specific dysregulated miRNAs in BAV patients.

We thank the Reviewer, and we discussed the recommended paper to further advance our Discussion on target genes and pathways linked to BAV and aneurysmal formation: Also, a pathway enrichment analysis from a miRNome profiling study in BAV aortopathy, identified a link with Hippo signaling pathway, ErbB signaling, TGF-beta signaling and focal adhesion, verifying the involvement of these pathways in the disease and their link with miRNA dysregulation (39).

Minor comments

11.- References 4 (line 55), 7 (line 58), 14-16 (line 273), and 20 (line 284) are not suitable. Also, eliminate reference 46.

The above-mentioned references have been eliminated from the text.

12.- Display BSA in line 105. However, this variable is not used in the whole study so, maybe it should be removed.

BSA abbreviation has been displayed in the text as body surface area. This parameter has been reported for completeness of the patient population description.

13.- Maybe ceramic beads are 2.8 mm? (line 114).

We would like to apologise for this typo, and it has been corrected to mm.

14.- Remove the first table with 9 patients in Table 1.

We would like to apologise for leaving an older version of the table, and it has been now removed.

15.- Also in Table 1, display SoV, N/A. There are no volunteers (V) in Table 1.

We thank the Reviewer for identifying the above. The SoV has been displayed as sinus of Valsalva, and the N/A as non-applicable. Also, the reference to volunteers has been removed from the table caption as no volunteers are shown in this paper.

16.- In Table 4, what does the p-values correspond to and what is the p<0.05 in the footnote? In addition, remove “as shown in Figure 1”, which represents the map of the ascending aorta segmentation.

The p-values in Tables 4 & 5, indicate the significance of the extracted Geneontology and protein pathways from Webgestalt for the predicted gene targets. In other words, a p-value of <0.05 indicates a significantly derived pathway to the corresponding gene target.

The “as shown in Figure 1” has been removed from both Table captions, and we apologise for the mistake.

17.- Change TGER3 by PTGER3 in Table 5. Separate the footnote from the main text (line 220).

We thank the Reviewer for spotting the above and have now been corrected.

18.- In Figure 4, check the size of the bar in A, with 200X magnification. It has to be less than in B.

We would like to apologise for this mistake. The scale bars have been corrected in the reviewed version of the figure.

In addition to the comments, small adjustments have been done, like inverting 2 paragraphs in Methods (Bioinformatic Analysis with Histomorphometric Analysis) and adding similar subheading to the Methods and Results to improve the paper’s reading-flow. A word document of the manuscript with highlighted changes (emailed to the Editorial office to be send to the Reviewers). The manuscript has now been transferred to the Cells online template with no highlighted changes as requested by the Editorial office. Also, a graphical abstract, updated figures and tables, and supplementary material will be emailed to the Editorial office to be send to the Reviewers.  

Round 2

Reviewer 2 Report

The authors have addressed most of my concerns satisfactorily but still some issues remain. I will keep the same numbering than in previous revision.

1. With the new explanation regarding the selection of the aortic segments for the analysis, it would seem correct the selection of A+AR vs. P+PL to assure the full spectrum of the dilated area, also considering the human error in the process of segmentation. However, the conflict still persists since the initial NGS and mass-spectrometry analyses were performed in AR vs. PL. Are differentially expressed miRNAs being lost by not considering those segments?

5. All the 92 miRNAs have FDR<1. Several times in both the manuscript and the reviewers’ responses you mention this threshold but I can guess that you must mean FDR<0.1. In any case, Table 2 is redundant with the supplementary Table with the 92 miRNAs, even when some values differ (see for example dilated vs. non-dilated-FC for hsa-miR-21-5p). Instead, a more informative Table 3 should include miRCURY LNA results from the 11 selected miRNAs.

Regarding miRNA data analysis, Methods section should refer to the supplementary appendix where a better description of the procedure should be provided. Current explanation is confusing (maybe an accurate reference should be more informative) and, in addition, the normalizers should be described.

6. Current Table 3 is not yet informative. First, is just p-value<0.05 in any comparison the one validating targets? Or, is there also any criteria such as FC>1.5 or Log2FC>0.5 (and FDR<0.1?) to validate the miRNAs? Must all four be met? Maybe those are too many conditions. It seems that only miR-150-5p is fulfilling the conditions. Thus, there is not a clear rule for the selection of miRNAs. Why it is stated that miR-210-3p was borderline (line 226) and later on, it was considered in a correlation with WSS (Figure 6)? All these criteria should be clarified in Methods for an accurate workflow. In addition, the inclusion of the “overall” patients in these results is not comment in text.

I apologize because I did not explain myself well in the previous comments regarding this table. Bold, if any, should highlight only significant values or values that meet the criteria described at the footnote.

As I have mentioned above, Table 3 should include miRCURY LNA results from the 11 selected miRNAs.

8. The explanation regarding Figure 3 (“we show that only 8 out of the 12 proteins are differentially expressed”) is still confusing because the initial 579 proteins already were differentially expressed proteins in the mass spectrometric analysis. Maybe you are adding in this figure an additional condition that is -0.5>log2(fold change)>0.5. Again, the criteria for the selection should be clarified.

The volcano plot with the 579 proteins looks nice and could be improved by removing the proteins inside the same cutoff values as in Figure 3.

11. Previous reference 7 was unsuitable in current line 77 but a reference is needed there.

Author Response

We Thank the Reviewers for the opportunity to clarify and improve our manuscript. The file with the replies to the Reviewers, the revised manuscript, and the updated figures have been uploaded. 
